Among-site variability in the stochastic dynamics of East African coral reefs

Allen Katherine A. 1 2
Bruno John F. 3
Chong Fiona 1
Clancy Damian 4
McClanahan Tim R. 5
Spencer Matthew m.spencer@liverpool.ac.uk 1
Żychaluk Kamila 6
1 School of Environmental Sciences, University of Liverpool , Liverpool , United Kingdom
2 Institute of Integrative Biology, University of Liverpool , Liverpool , United Kingdom
3 Department of Biology, University of North Carolina at Chapel Hill , Chapel Hill , NC , United States of America
4 School of Mathematical and Computer Sciences, Actuarial Mathematics and Statistics, Heriot-Watt University , Edinburgh , United Kingdom
5 Wildlife Conservation Society , NY , United States of America
6 Department of Mathematical Sciences, University of Liverpool , Liverpool , United Kingdom
Hay Mark
Electronic publication date: 2017 May 17
Publication date: 2017
Volume: 5
Electronic Location ID: e3290
Received 2017 Jan 3; Accepted 2017 Apr 10
Copyright: ©2017 Allen et al.
Copyright year: 2017
Copyright holder: Allen et al.
License: This is an open access article distributed under the terms of the Creative Commons Attribution License, which permits unrestricted use, distribution, reproduction and adaptation in any medium and for any purpose provided that it is properly attributed. For attribution, the original author(s), title, publication source (PeerJ) and either DOI or URL of the article must be cited.
License URL: https://creativecommons.org/licenses/by/4.0/

Keywords: Vector autoregressive model, State-space model, Stochastic dynamics, Community composition, Spatial variability, Temporal variability, Coral reef, Bayesian statistics

Funding: NERC NE/K00297X/1 This work was supported by NERC grant NE/K00297X/1. The funders had no role in study design, data collection and analysis, decision to publish, or preparation of the manuscript.

==============================
Coral reefs are dynamic systems whose composition is highly influenced by unpredictable biotic and abiotic factors. Understanding the spatial scale at which long-term predictions of reef composition can be made will be crucial for guiding conservation efforts. Using a 22-year time series of benthic composition data from 20 reefs on the Kenyan and Tanzanian coast, we developed Bayesian vector autoregressive state-space models for reef dynamics, incorporating among-site variability, and quantified their long-term behaviour. We estimated that if there were no among-site variability, the total long-term variability would be approximately one-third of its current value. Thus, our results showed that among-site variability contributes more to long-term variability in reef composition than does temporal variability. Individual sites were more predictable than previously thought, and predictions based on current snapshots are informative about long-term properties. Our approach allowed us to identify a subset of possible climate refugia sites with high conservation value, where the long-term probability of coral cover ≤0.1 (as a proportion of benthic cover of hard substrate) was very low. Analytical results show that this probability is most strongly influenced by among-site variability and by interactions among benthic components within sites. These findings suggest that conservation initiatives might be successful at the site scale as well as the regional scale.

Introduction

“Probabilistic language based on stochastic models of population growth” has been proposed as a standard way to evaluate conservation and management strategies (Ginzburg et al., 1982). For example, a stochastic population model can be used to estimate the probability of abundance falling below some critical level. Such population viability analyses are widely used, and may be reasonably accurate if sufficient data are available (Brook et al., 2000). In principle, the same approach could be used for communities, provided that a sufficiently simple model of community dynamics can be found.

A good candidate for such a model is the vector autoregressive model of order 1 or VAR(1) (Lütkepohl, 1993; Ives et al., 2003). This is a discrete-time model for the vector of log abundances of a set of species or groups, which includes environmental stochasticity and may include environmental explanatory variables. It makes the simplifying assumptions that inter- and intraspecific interactions can be represented by a linear approximation on the log scale, and that future abundances are conditionally independent of past abundances, given current abundances. Where possible, it is desirable to use a state-space form of the VAR(1) model, which also includes measurement error (Lindegren et al., 2009; Mutshinda, O’Hara & Woiwod, 2009).

Hampton et al. (2013) review applications of VAR(1) models in community ecology, which include studying the stability of freshwater plankton systems (Ives et al., 2003), designing adaptive management strategies for the Baltic Sea cod fishery (Lindegren et al., 2009), and estimating the contributions of environmental stochasticity and species interactions to temporal fluctuations in abundance of moths, fish, crustaceans, birds and rodents (Mutshinda, O’Hara & Woiwod, 2009). Recently, VAR(1) models have been applied to the dynamics of the benthic composition of coral reefs (Cooper, Spencer & Bruno, 2015; Gross & Edmunds, 2015), using a log-ratio transformation (Egozcue et al., 2003) rather than a log transformation, to deal with the constraint that proportional cover of space-filling benthic groups sums to 1.

Coral reefs are dynamic systems influenced by both deterministic factors such as interactions between macroalgae and hard corals (Mumby, Hastings & Edwards, 2007), and stochastic factors such as temperature fluctuations (Baker, Glynn & Riegl, 2008) and storms (Connell, Hughes & Wallace, 1997), and are classic examples of non-equilibrium systems whose diversity is determined by both interspecific interactions and disturbance (Huston, 1985). In general, high coral cover is considered a desirable state for a coral reef, and there is some evidence that coral cover of at least 0.1 (as a proportion of substrate, equivalent to 10%) is important for long-term maintenance of reef function (Kennedy et al., 2013; Perry et al., 2013; Perry et al., 2015; Roff, Zhao & Mumby, 2015). A positive net carbonate budget is necessary to maintain reef functions such as provision of habitat (Kennedy et al., 2013). Simulation models of Caribbean reefs suggested that coral cover of 0.1 was just sufficient to maintain a zero net carbonate budget under a scenario with low greenhouse gas emissions and protection of herbivorous fish (Kennedy et al., 2013). Statistical analysis of field data from Caribbean reefs (Perry et al., 2013) supported the idea that coral cover greater than 0.1 is required for a positive net carbonate budget, and a model of bioerosion parameterized using field data on Orbicella annularis-dominated reefs in Belize suggested a similar coral cover threshold of 0.05–0.1 (Roff, Zhao & Mumby, 2015). Statistical analysis of field data from the Chagos Archipelago also supported a coral cover threshold of 0.1 for a positive net carbonate budget on reefs dominated by Porites and Pocillopora, although the threshold was lower for sites dominated by Acropora (Perry et al., 2015). Thus, overall, coral cover of 0.1 might be an appropriate threshold against which to evaluate reef conservation strategies, and VAR(1) models can be used to estimate the probability of coral cover falling to or below this threshold (Cooper, Spencer & Bruno, 2015).

There is evidence for systematic differences in reef dynamics among locations. For example, on the Great Barrier Reef up to 2012, coral cover had declined more strongly at southern and central than at northern sites (De’ath et al., 2012), and in the US. Virgin Islands, VAR(1) models showed that sites differed in their sensitivity to disturbance and speed of recovery (Gross & Edmunds, 2015). Some sites in a region may therefore represent coral refugia, where reefs are either protected from or able to adapt to changes in environmental conditions (McClanahan et al., 2007b). Alternatively, apparent differences among sites may simply be due to differences in recent acute disturbance history, and may not persist in the long term (e.g., Connell, 1997). Although it may be possible to associate differences in dynamics among sites with differences in environmental variables, it is also possible to treat among-site differences as another random component of a VAR(1) model. This will allow estimation of the relative importance of among-site variability and within-site temporal variability, which is important for the design of conservation strategies. If within-site temporal variability dominates, it will not be possible to identify good sites to conserve based on current status, while if among-site variability dominates, even a “snapshot” sample at one time point may be enough to identify good sites. Thus, for example, the reliability of among-site patterns from surveys at one time point, such as the relationship between benthic composition and human impacts on remote Pacific atolls (Sandin et al., 2008), depends on among-site variability dominating within-site temporal variability. Thus, even though a simple strategy based on a snapshot may turn out to be effective, it is not possible to know this in advance of carrying out a more sophisticated analysis that treats the system as dynamic. As far as we know, the use of VAR(1) models to estimate spatiotemporal heterogeneity and identify refugia is novel, although other applications of VAR(1) models with random subject effects exist (e.g. Gorrostieta et al., 2012; Driver, Oud & Voelkle, 2016). Our approach differs from existing methods for identifying refugia (Keppel et al., 2012) in that it explicitly focuses on spatial variability in dynamics over ecological timescales, rather than on patterns that are static or vary only over much longer timescales. Furthermore, rather than differences in physical factors (West & Salm, 2003), we focus on differences in community dynamics.

Here, we develop a state-space VAR(1) model for regional dynamics of East African coral reefs, including random site effects and measurement error, and use it to answer four key questions about spatial and temporal variability. How important is among-site variability in the dynamics of benthic composition, relative to within-site temporal variability? How much variability is there among sites in the probability of low (≤0.1) coral cover? Which model parameters have the largest effects on the probability of low coral cover in the region? How informative is a single snapshot in time about the long-term properties of a site?

Methods

Data collection

Surveys of 20 spatially distinct reefs in Kenya and Tanzania (Table 1 and Fig. A1) were conducted annually during the period 1991–2013 (generally in November or December prior to 1998, but January or February from 1998 onwards). Sampling dates are shown in Fig. A6–A35. Reefs in the north were typically fringing reefs, 100 m–2,000 m from the shore, while those in the south were typically smaller and more isolated patch reefs, further from the shore (McClanahan & Arthur, 2001). We categorized reefs as either fished or unfished, although there was substantial heterogeneity within these categories, because some fished reefs were community management areas with reduced harvesting intensity (Cinner & McClanahan, 2015), and some unfished reefs had only recently been designated as reserves. Of the 20 reefs, 10 were divided into two sites separated by 20 m–100 m, while the remaining 10 reefs comprised only one site. The selection of sites represents available data rather than a random sample from all the locations at which coral reefs are present in the geographical area (and all of the longest time series are from Kenyan fringing reefs). Below, we use a random term to model the variation in dynamics among these sites. Thus, when we refer to ‘a randomly-chosen site’ we mean ‘a site drawn at random from the distribution describing variability in dynamics among our sites’, which is not the same as a site drawn at random from all coral reef locations in the region.

Table 1 Sampling information and reef features.

For each named reef, surveys were done at either one site, or at two sites 20 m–100 m apart. Fished reefs include community management areas with reduced harvesting intensity, and unfished reefs include those recently designated as reserves. Mean coral cover is the arithmetic mean of observed coral cover over all transects and time points.

Reef	Sites	Location	Time points	Time range	Reef type	Management	Mean coral cover (site 1, site 2)	
Bongoyo	2	6.67S, 39.26E	3	1995–2012	patch	fished	54.7, 52.1	
Changale	1	5.30S, 39.10E	3	1995–2010	patch	fished	39.4	
Changuu	1	6.12S, 39.12E	3	1997–2012	patch	fished	46.8	
Chapwani	1	6.07S, 39.11E	3	1997–2012	patch	fished	52.5	
Chumbe	2	6.28S, 39.17E	3	1997–2012	patch	unfished	70.1, 74.1	
Diani	2	4.37S, 39.58E	19 (site 1), 18 (site 2)	1992–2013	fringing	fished	32.0, 17.5	
Funguni	1	5.27S, 39.13E	3	1995–2010	patch	fished	13.7	
Kanamai	2	3.93S, 39.78E	19	1991–2013	fringing	fished	33.0, 32.3	
Kisite	2	4.71S, 39.37E	8 (site 1), 9 (site 2)	1994–2012	patch	unfished	33.9, 46.4	
Makome	1	5.28S, 39.11E	3	1995–2010	patch	fished	32.1	
Malindi	2	3.26S, 40.15E	20	1991–2013	fringing	unfished	27.9	
Mbudya	2	6.66S, 39.25E	3	1995–2012	patch	fished	53.5, 68.0	
Mombasa	2	3.99S, 39.75E	20	1991–2013	fringing	unfished	37.27, 29.2	
Mradi	1	3.94S, 39.78E	2	2010–2011	fringing	fished	48.4	
Nyali	2	4.05S, 39.71E	2	2006–2009	fringing	fished	28.1, 29.1	
Ras Iwatine	1	4.02S, 39.73E	18	1993–2013	fringing	fished	10.8	
Taa	1	3.99S, 39.77E	3	1995–2010	patch	fished	20.7	
Tiwi Inside	1	4.26S, 39.61E	2	2008–2011	fringing	fished	36.0	
Vipingo	2	3.48S, 39.95E	18 (site 1), 17 (site 2)	1991–2013	fringing	fished	28.0, 28.2	
Watamu	1	3.37S, 40.01E	20	1991–2013	fringing	unfished	23.2	

Each of the 30 sites was visited at least twice (data from sites visited once were omitted), with a maximum of 20 visits. In total, there were 289 site visits. In each, a version of line-intercept sampling (Kaiser, 1983; McClanahan, Muthiga & Mangi, 2001) was used to estimate reef composition. In total, 2,665 linear transects were sampled across all sites and years, with between five and 18 transects (median 9) at each site in a single year. Transects were randomly placed between two points 10 m apart, but as the transect line was draped over the contours of the substrate, the measured lengths varied between 10 m and 15 m. Cover of benthic taxa was recorded as the sum of draped lengths of intersections of patches of each taxon with the line, divided by the total draped length of the line. Intersections with length less than 3 cm were not recorded. Taxa were identified to species or genus level, but for this study cover was grouped into three broad categories: hard coral (scleractinians and Millepora), macroalgae and other (algal turf, calcareous and coralline algae, soft corals and sponges). Millepora were included in the coral category because they are important calcareous framework builders and may have calcification rates similar to those of branching corals (Lewis, 1989). Sand and seagrass were recorded, but excluded from our analysis, which focused on hard substrate. The dynamics of a subset of these data were analyzed using different methods in Żychaluk et al. (2012).

Data processing

The three cover values form a three-part composition, a set of three positive numbers whose sum is 1 (Aitchison, 1986, Definition 2.1, p. 26). Standard multivariate statistical techniques are not appropriate for untransformed compositional data, due to the absence of an interpretable covariance structure and the difficulties with parametric modelling (Aitchison, 1986, chapter 3). To avoid these difficulties, the proportional cover data were transformed to orthogonal, unconstrained, isometric log-ratio (ilr) coordinates (Egozcue et al., 2003). It is of course true that the model presented below for transformed data has an analogous model for untransformed data (Mateu-Figueras, Pawlowsky-Glahn & Egozcue, 2011). However, working with transformed data allows us to use familiar methods.

The transformed data at site i, transect j, time t were represented by the vector yi,j,t = [y1,i,j,t, y2,i,j,t]T, in which the first coordinate y1,i,j,t was proportional to the natural log of the ratio of algae to coral, and the second coordinate y2,i,j,t was proportional to the natural log of the ratio of other to the geometric mean of algae and coral (Section A1). The T denotes transpose: throughout, we work with column vectors. Note that both raw and transformed data are dimensionless.

The model

The true value xi,t = [x1,i,t, x2,i,t]T of the isometric log-ratio transformation of cover of hard corals, macroalgae and other at site i at time t was modelled by a vector autoregressive process of order 1 (i.e., a process in which the cover in a given year depends only on cover in the previous year), an approach used in other recent models of coral reef dynamics (Cooper, Spencer & Bruno, 2015; Gross & Edmunds, 2015). Unlike previous models, we include a random term representing among-site variation, and explicit treatment of measurement error (making this a state-space model). The full model is (1) xi,t+1=a+αi+Bxi,t+εi,t,αi∼N0,Z,εi,t∼N0,Σ,yi,j,t∼t2xi,t,H,ν,

where all variables and parameters are dimensionless.

The column vector a represents the among-site mean proportional changes in xi,t evaluated at xi,t = 0. The column vector αi represents the amount by which these proportional changes for the ith site differ from the among-site mean, and is assumed to be drawn from a multivariate normal distribution with mean vector 0 and 2 × 2 covariance matrix Z. The 2 × 2 matrix B represents the effects of xi,t on the proportional changes, and can be thought of as summarizing intra- and inter-component interactions such as competition. The column vector εi,t represents random temporal variation, and is assumed to be drawn from a multivariate normal distribution with mean vector 0 and covariance matrix Σ. We assume that there is no temporal or spatial autocorrelation in ε, and that ε is independent of the among-site variation α.

The observed transformed compositions yi,j,t vary around the corresponding true compositions xi,t due to both small-scale spatial variation in true composition among transects within a site, and measurement error in estimating composition from a transect. We cannot easily separate these sources of variation because transects were located at different positions in each year, and there were no repeat measurements within transects. Observed log-ratio transformed cover yi,j,t in the jth transect of site i at time t was assumed to be drawn from a bivariate t distribution (denoted by t2) with location vector equal to the corresponding xi,t, and unknown scale matrix H and degrees of freedom ν, so that yi,j,t has mean vector xi,t if ν > 1, and covariance matrix νH∕(ν − 2) if ν > 2 (Lange, Little & Taylor, 1989). The bivariate t distribution can be interpreted as a mixture of bivariate normal distributions whose covariance matrices are the same up to a scalar multiple (Lange, Little & Taylor, 1989), and therefore allows a simple form of among-site or temporal variation in the distribution of measurement error or small-scale spatial variation, whose importance increases as the degrees of freedom decrease. Preliminary analyses suggested that it was important to allow this variation, because the model in Eq. (1) fitted the data much better than a model with a bivariate normal distribution for yi,j,t (Section A3).

We make the important simplifying assumptions that B is the same for all sites, and that the causes of among-site and temporal variation are not of interest. A separate B for each site, or even a hierarchical model for B, would be difficult to estimate from the amount of data we have. It might be possible to explain some of the random temporal variation using temporally-varying environmental covariates such as sea surface temperature, and some of the among-site variation using temporally constant covariates such as management strategies (Cooper, Spencer & Bruno, 2015). However, it is not necessary to do so in order to answer the questions listed at the end of the introduction, and keeping the model as simple as possible is important because parameter estimation is quite difficult. Furthermore, some of the relevant environmental variables may be associated with management strategies, making it difficult to separate the effects of environmental variation and management. For example, although some water quality variables were not strongly associated with protection status (Carreiro-Silva & McClanahan, 2012), unfished reefs were designated as protected areas due to their relatively good condition and are generally found in deeper lagoons with lower and more stable water temperatures than fished reefs (TR McClanahan, pers. comm., 2015).

To understand the features of dynamics common to all sites, we plotted the back-transformations from ilr coordinates to the simplex of the overall intercept parameter a and the columns a1 and a2 of a matrix A, which is related to B and describes the effects of current reef composition on the change in reef composition from year to year (Cooper, Spencer & Bruno, 2015). We plotted A rather than B because it leads to a simpler visualization of effects (Section A4). For example, a point lying to the left of the line representing equal proportions of coral and algae (the 1:1 coral-algae isoproportion line) corresponds to a parameter tending to increase coral relative to algae.

Parameter estimation

We estimated all model parameters and checked model performance using Bayesian methods implemented in the Stan programming language (Stan Development Team, 2015), as described in Section A5. Stan uses the No-U-Turn Sampler, a version of Hamiltonian Monte Carlo, which can converge much faster than random-walk Metropolis sampling when parameters are correlated (Hoffman & Gelman, 2014). For most results, we report posterior means and 95% highest posterior density (HPD) intervals (Hyndman, 1996), calculated in R (R Core Team, 2015). We showed using simulations that we were able to estimate parameters with reasonable accuracy, and that our estimated credible intervals had close to the specified coverage (Section A5.2 and Fig. A3). In order to investigate the effects of number of time points per site on uncertainty in site-specific parameters, we plotted the sample generalized variance of αi (the determinant of the sample covariance matrix over Monte Carlo iterations) against number of time points. It is worth emphasizing that all parameters other than αi and xi,t are common to all reefs, and thus estimated from all 289 site visits.

Long-term behaviour

In the long term (as t → ∞), the true transformed composition x∗ of a randomly-chosen site will converge to a stationary distribution, provided that all the eigenvalues of B lie inside the unit circle in the complex plane (e.g. Lütkepohl, 1993, p. 10). If the eigenvalues of B are complex, the system will oscillate as it approaches the stationary distribution. Details of long-term behaviour are in Section A6.

This stationary distribution is the multivariate normal vector (2) x∗∼Nμ∗,Σ∗+Z∗,

whose stationary mean μ∗ depends on B and a, and whose stationary covariance is the sum of the stationary within-site covariance Σ∗ (which depends on B and Σ) and the stationary among-site covariance Z∗ (which depends on B and Z).

For a fixed site i, the value of αi is fixed and the stationary distribution is given by (3) xi∗∼Nμi∗,Σ∗,

whose stationary mean μi∗ depends on B, a and αi, and whose stationary covariance matrix is Σ∗. Note that B, which describes intra- and inter-component interactions on an annual time scale, affects all the parameters of both stationary distributions, and therefore affects both within- and among-site variability in the long term. Also, the back-transformation of the stationary mean μ∗ of the transformed composition, rather than the arithmetic mean vector of the untransformed composition, is the appropriate measure of the centre of the stationary distribution (Aitchison, 1989).

How important is among-site variability?

The covariance matrix of the stationary distribution for a randomly-chosen site (Eq. (2)) contains contributions from both among- and within-site variability. To quantify the contributions from these two sources, we calculated (4) ρ=|Σ∗||Σ∗+Z∗|1∕2,

(Section A7), which is the ratio of volumes of two unit ellipsoids of concentration (Kenward, 1979), the numerator corresponding to the stationary distribution in the absence of among-site variation (or for a fixed site, as in Eq. (3)), and the denominator to the full stationary distribution of transformed reef composition in the region. The volume of each ellipsoid of concentration is a measure of the dispersion of the corresponding distribution. Thus ρ provides an indication of how much of the total variability would remain if all among-site variability was removed. A similar statistic was used by Ives et al. (2003) to measure the contribution of species interactions to stationary variability.

How much variability is there among sites in the probability of low coral cover?

For a given coral cover threshold κ, we define qκ,i as the long-term probability that site i has coral cover less than or equal to κ. This can be interpreted either as the proportion of time for which the site will have coral cover less than or equal to κ in the long term, or as the probability that the site will have coral cover less than or equal to κ at a random time, in the long term. We set κ = 0.1, which has been suggested as a threshold for a positive net carbonate budget, based on simulation models and data from Caribbean reefs (Kennedy et al., 2013; Perry et al., 2013; Roff, Zhao & Mumby, 2015). We calculated q0.1,i for each site numerically (Section A8). In order to determine whether differences in q0.1,i were related to current coral cover, we plotted q0.1,i against the corresponding sample mean coral cover for each site, over all transects and years. In order to determine whether differences in q0.1,i had obvious explanations, we distinguished between fished and unfished reefs, and patch and fringing reefs.

In order to determine how the amount of among-site variability affects the strength of the relationship between q0.1,i and sample mean coral cover, we plotted this relationship for simulated data sets with different amounts of among-site variability (Section A9). In order to determine whether there was strong spatial pattern in the probability of low coral cover, we calculated spline correlograms (Bjørnstad & Falck, 2001) for a sample from the posterior distribution of q0.1,i (Section A10).

Which model parameters have the largest effects on the probability of low coral cover?

For a given coral cover threshold κ, we define qκ as the long-term probability that a randomly-chosen site has coral cover less than or equal to κ. This is equal to the expected long-term probability that coral cover is less than or equal to κ over the region, and can be calculated numerically (Section A8). To find the parameters with the largest effects on qκ, we calculated its derivatives with respect to each model parameter. As above, we concentrated on κ = 0.1. However, we also compared results from κ = 0.05 and κ = 0.20. The probability qκ is a function of 12 parameters: all four elements of B; both elements of a; elements σ11, σ21 and σ22 of Σ; and elements ζ11, ζ21 and ζ22 of Z. The negative of the gradient vector of derivatives of qκ with respect to these parameters describes the direction of movement through parameter space in which the probability of low coral cover will be reduced most rapidly, and the elements of this vector with the largest magnitudes correspond to the parameters to which qκ is most sensitive. To understand why qκ responds to each model parameter, note that qκ depends on the parameters μ∗, Σ∗ and Z∗ of the stationary distribution (Eq. (2)), which are in turn affected by the model parameters. We therefore used the chain rule for matrix derivatives (Magnus & Neudecker, 2007, p.108) to break down the derivatives into effects of μ∗, Σ∗ and Z∗ on qκ, and effects of model parameters on μ∗, Σ∗ and Z∗ (Section A11). We also calculated elasticities of qκ with respect to each parameter, which measure the rate of relative change in qκ with respect to relative change in the parameter (Section A12).

How informative is a snapshot about long-term site properties?

In a stochastic system, how much can a “snapshot” survey at a single point in time tell us about the long-term behaviour of the system? For example, are differences among sites that appear to be in good and bad condition likely to be maintained in the long term? To make this question more precise, suppose that we draw a site at random from the region, and at one point in time, draw the true state of the site at random from the stationary distribution for the site. This scenario matches Diamond’s definition of “natural snapshot experiments” as “comparisons of communities assumed to have reached a quasi-steady state” (Diamond, 1986). For simplicity, we assume that we can estimate the true state accurately (for example, by taking a large number of transects). To quantify how informative this is about the long term properties of the site, we computed the correlation coefficients between corresponding components of the true state at a given site at a given time and of stationary mean for that site (Section A13). If these correlations are high, then a snapshot will be informative about long term properties.

Results

Overall dynamics

At all sites, the back-transformed posterior mean true states from the model (e.g., Fig. 1, grey lines) closely tracked the centres of the distributions of cover estimates from individual transects, although there was substantial among-transect variability at a given site in a given year (e.g., Fig. 1, circles). Figure 1 shows two examples, and time series for all sites are plotted in Figs. A6–A35. There were also substantial differences in patterns of temporal change among sites. For example, Kanamai1 (Figs. 1A–1C), a fished site, had consistently low algal cover and no dramatic changes in cover of any component. In contrast, Mombasa1 (Figs. 1D–1F), an unfished site, had a sudden decrease in coral cover in 1998, and algal cover was high from 2007 onwards. As a result, Mombasa1 was unusual in that the current estimate of true algal cover was well above the stationary mean estimate (Fig. 1E: black circle at end of time series). For most other sites, current estimated true cover was close to the stationary mean (Figs. A6–A35, black circles at ends of time series). The uncertainty in true states (Fig. 1, grey polygons represent 95% highest posterior density (HPD) credible bands) was higher during intervals with missing observations (e.g., 2008 in Fig. 1). Uncertainty in true states (grey polygons) and stationary means (black bars at end of time series) was high for sites with few observations (e.g., Bongoyo1, Fig. A6). This is expected because these quantities are estimated at the site level. Similarly, the generalized variance of αi was high for sites with only two or three observations, but declined quickly as the number of observations increased (Fig. A36).

Figure 1 Time series of cover of hard corals, macroalgae and other at two of the 30 sites surveyed: Kanamai1 (fished, A–C) and Mombasa1 (unfished, D–F).

Circles are observations from individual transects. Grey lines join back-transformed posterior mean true states from Eq. (1), and the shaded region is a 95% highest posterior density band. The back-transformed stationary mean composition for the site is the black dot after the time series and the bar is a 95% highest posterior density interval.

The overall intercept parameter a (Fig. 2, green), which describes the dynamics of reef composition at the origin (where each component is equally abundant) was consistent with the observed low macroalgal cover in the region (e.g., Figs. 1B and 1E). The back-transformation of a lay close to the coral-other edge of the ternary plot, and slightly above the 1:1 coral-other isoproportion line. It therefore represented a strong year-to-year decrease in algae, and a slight increase in other relative to coral, at the origin.

Figure 2 Posterior distributions of the back-transformed overall intercept a (green), effect a1 of component 1 (proportional to log(algae/coral)) on year-to-year change (orange), and effect a2 of component 2 (proportional to log(other/geometric mean(algae,coral)) on year-to-year change (blue).

Current reef composition acts on year-to-year change in composition (through matrix A) so as to maintain fairly stable reef composition, with an increase in each transformed component tending to be counteracted in the following year-to-year change. Each column of A represents the effect of a unit increase in a component of transformed composition on the year-to-year change in transformed composition (Section A4). These effects can be back-transformed to compositions and plotted on a ternary diagram (Fig. 2). The effects can be interpreted by looking at the position of the resulting point relative to three 1:1 isoproportion lines (Fig. 2, grey lines), along each of which the relative proportions of two components do not change: the 1:1 coral-algae isoproportion line, representing no change in coral relative to algae (from the the middle of the coral-algae edge to the other vertex); the 1:1 algae-other isoproportion line, representing no change in algae relative to other (from the middle of the algae-other edge to the coral vertex); and the 1:1 coral-other isoproportion line, representing no change in coral relative to other (from the middle of the coral-other edge to the algae vertex).

The back-transformation of the first column a1 of A, which represents the effects of the transformed ratio of algae to coral on year-to-year change in composition, lay to the left of the 1:1 coral-algae isoproportion line, above the 1:1 other-algae isoproportion line, and below the 1:1 coral-other isoproportion line (Fig. 2, orange). Thus, increases in algae relative to coral resulted in decreases in algae relative to coral and other, and increases in coral relative to other, in the following year. The back-transformation of the second column a2 of A, which represents the effects of the transformed ratio of other to algae and coral on year-to-year change in composition, lay on the 1:1 coral-algae isoproportion line, below the 1:1 other-algae isoproportion line, and below the 1:1 coral-other isoproportion line (Fig. 2, blue). Thus, increases in other relative to algae and coral resulted in little change in the ratio of coral to algae, but decreases in other relative to both coral and algae. Consistent with this evidence of a tendency towards stability in year-to-year dynamics, every set of parameters in the Monte Carlo sample led to a stationary distribution, since both eigenvalues of B lay inside the unit circle in the complex plane (Section A14). The magnitudes of these eigenvalues were smaller than those for a similar model for the Great Barrier Reef (Cooper, Spencer & Bruno, 2015), indicating more rapid approach to the stationary distribution. There was some evidence for complex eigenvalues of B, leading to rapidly-decaying oscillations in both components of transformed reef composition on approach to this distribution. This contrasts with the Great Barrier Reef, where there was no evidence for oscillations (Cooper, Spencer & Bruno, 2015).

In biological terms, the above results mean that every site would, if current environmental conditions were maintained in the long term, approach a predictable probability distribution of composition, whatever the initial conditions. However, as described below, these distributions differed substantially among sites.

How important is among-site variability?

There was substantial among-site variability in the locations of stationary means (Fig. 3, dispersion of points). Stationary mean algal cover was always low, but there was a wide range of stationary mean coral cover. Although our primary focus is not on the causes of among-site variability, there was a tendency for most of the reefs with highest stationary mean coral cover to be patch reefs (Figs. 3A and 3C). In the light of these observations, we experimented with a model in which reef type was included as an explanatory variable. Although the estimated effects of patch reef type were consistent with lower long-term probabilities of coral cover ≤0.1, including reef type did not improve the expected predictive accuracy of the model (Chong, 2016), probably because only 482 out of 2,665 transects were from patch reefs, and all but one patch reefs had only very short time series (Table 1). The stationary means did not clearly separate by management (Figs. 3A and 3B versus Figs. 3C and 3D). The long-term temporal variability around the stationary means was also substantial (Fig. 3, green lines). The ρ statistic (Eq. (4)), which quantifies the posterior mean contribution of within-site variability to the total stationary variability in reef composition in the region, was 0.29 (95% HPD interval (0.20, 0.39)), or approximately one-third. Thus, while within-site temporal variability around the stationary mean was not negligible, among-site variability in the stationary mean was more important in the long term. As noted above, uncertainty in the location of the stationary means (Fig. 3, grey dashed lines) was much higher for reefs with few observations than for reefs with many observations. Nevertheless, most parameters of the model are not reef-specific, and data from reefs with few observations contribute to the estimation of these.

Figure 3 Stationary among- and within-site variation in benthic composition on (A) fished patch reefs, (B) fished fringing reefs, (C) unfished patch reefs, and (D) unfished fringing reefs.

Grey points: back-transformed posterior means of stationary means for each site. Grey dashed curves: back-transformed unit ellipsoids of concentration representing uncertainty in stationary means (calculated using sample covariance matrices from Monte Carlo iterations). Green solid curves: back-transformed unit ellipsoids of concentration representing within-site stationary variation (calculated using posterior mean within-site covariance matrix).

For all three components of variability (within-site, among-site, and measurement error/small-scale spatial variability), variation in algal cover was larger than variation in coral or other. This can be seen in the shapes of the back-transformed unit ellipsoids of concentration (Fig. 4: within-site, green; among-site, orange; measurement error and small-scale spatial variability, blue) which were all elongated to some extent along the 1:1 coral-other isoproportion line. This was similar to, but less extreme than, the pattern observed in the Great Barrier Reef (Cooper, Spencer & Bruno, 2015). The among-site ellipsoid almost entirely enclosed the within-site ellipsoid, consistent with the estimate above that within-site variability contributed only around one-third of the total stationary variability in reef composition. The large estimated measurement error/small-scale spatial variability component was consistent with the substantial observed variability in cover among transects at any given site and time (Fig. 1, circles and Figs. A6–A35, circles). The low estimated degrees of freedom ν for the bivariate t distribution of measurement error/small-scale spatial variability (posterior mean 2.99, 95% HPD interval (2.64, 3.35)) suggested that some aspect of the process leading to variation in measured composition among transects at a given site was varying substantially over space or time, although we cannot determine the mechanism.

Figure 4 Back-transformed unit ellipsoids of concentration for stationary within-site covariance Σ∗ (green), stationary among-site covariance Z∗ (orange), and measurement error/small-scale spatial variation νH∕(ν − 2) (blue).

In each case, 200 ellipsoids drawn from the posterior distribution are plotted, centred on the origin.

How much variability is there among sites in the probability of low coral cover?

There was also substantial among-site variability in the probability of low coral cover. For a randomly-chosen site, the posterior mean probability of coral cover less than or equal to 0.1 (q0.1) in the long term was 0.12 (95% credible interval (0.04, 0.21)). The corresponding site-specific probabilities q0.1,i varied from 8 × 10−5 to 0.52 but were low for most sites, with a strong negative relationship between probability of low coral cover and observed mean coral cover (Fig. 5).

Figure 5 Long-term probability of coral cover less than or equal to 0.1 at each site against mean observed coral cover across all years.

Circles are patch reefs and triangles are fringing reefs. Open symbols are fished reefs and shaded symbols are unfished. Vertical lines are 95% highest posterior density intervals.

There was no clear distinction in the probability of low coral cover between fished and unfished reefs (Fig. 5, open symbols fished, filled symbols unfished). However, probability of low coral cover appeared to be systematically lower on patch reefs, which were mainly in Tanzania (Figs. 5 and A1, circles: median of posterior means 2 × 10−3, first quartile 4 × 10−4, third quartile 0.04) than on fringing reefs (Figs. 5 and A1, triangles: median of posterior means 0.08, first quartile 0.04, third quartile 0.11). One site (Ras Iwatine) had a much higher probability of low coral cover than all others, and is one of two relatively eutrophic sites (the other being Kanamai), probably due to pollution (Carreiro-Silva & McClanahan, 2012).

Figure 6 Elements of the gradient vector of partial derivatives of the long-term probability of coral cover less than or equal to 0.1 with respect to elements of the B matrix (effects of transformed composition in a given year on transformed composition in the following year), the a vector (overall intercept, representing among-site mean proportional changes in transformed composition at the origin), the covariance matrix of random temporal variation Σ, and the covariance matrix of among-site variability Z.

For each parameter, the dot is the posterior mean and the bar is a 95% highest posterior density credible interval. For the covariance matrices, the elements σ12 and ζ12 are not shown, because they are constrained to be equal to σ21 and ζ21 respectively. The horizontal dashed line is at zero, the no-effect value.

Although a negative relationship between the probability of low coral cover and observed mean coral cover (Fig. 5) was expected, the strength of this relationship depends on the amount of among-site variability. Using simulated data, we showed that when there was much less among-site variability than estimated from the real data, this relationship was very weak, and the probability of low coral cover was small for all sites (Fig. A39). As the amount of among-site variability increased, the probability of low coral cover increased quickly for sites with low mean coral cover, but remained close to zero for sites with high mean coral cover.

There was little evidence for strong spatial autocorrelation in the probability of low coral cover, because the 95% envelope for the spline correlogram included zero for all distances other than 261 km to 322 km (Fig. A40). The general lack of strong spatial autocorrelation reflects the substantial variation in probability of coral cover less than or equal to 0.1 (q0.1,i) among nearby sites, while the possibility of negative spatial autocorrelation at scales of around 300 km may reflect the generally low values of q0.1,i for Tanzanian patch reefs, separated from sites in the north of the study area with generally higher q0.1,i by approximately 300km (Fig. A1).

Which model parameters have the largest effects on the probability of low coral cover?

Both among-site variability and internal dynamics, particularly of other relative to algae and coral (component 2), were important in determining the probability q0.1 of coral cover ≤0.1 in the region. Figure 6 shows the direction in parameter space along which the probability of low coral cover will reduce most rapidly (the estimated gradient vector of q0.1 with respect to all the model parameters). The four parameters to which q0.1 was most sensitive were (in descending order: Fig. 6) ζ21 (among-site covariance between transformed components 1 and 2), b22 (effect of component 2 on next year’s component 2), ζ22 (among-site variance of component 2), and b12 (effect of component 2 on next year’s component 1). Although there was substantial variability among Monte Carlo iterations in the values of these derivatives, the rank order of magnitudes was fairly consistent (Fig. A41). All four most important parameters had positive effects on q0.1 (Fig. 6), so reducing these parameters will reduce q0.1. The effects of within-site temporal variability on the probability of low coral cover were relatively unimportant (Fig. 6, derivatives of q0.1 with respect to σ11, σ21 and σ22 all had posterior means close to zero). The signs of the effects of each parameter on q0.1, sensitivities for coral cover thresholds 0.05 and 0.2, and elasticities, are discussed further in Sections A15 and A16.

How informative is a snapshot about long-term site properties?

For both components of transformed composition, a snapshot of reef composition at a single time on a randomly-chosen site will be informative about the stationary mean (correlations between true value at a given time and stationary mean: component 1 posterior mean 0.84, 95% HPD interval (0.75, 0.91); component 2 posterior mean 0.82, 95% HPD interval (0.73, 0.90)). This is consistent with the negative relationship between long-term probability of coral cover ≤0.1 and observed mean coral cover (Fig. 5). Thus, while long-term monitoring of East African coral reefs is important for other reasons, it should be possible to identify those with high conservation value (in terms of benthic composition) from a single survey.

Discussion

In the long term (as t → ∞), among-site variability dominated within-site temporal variability in East African coral reefs. In consequence, the long-term probability of coral cover ≤0.1 varied substantially among sites. This suggests that it is in principle possible to make reliable decisions about the conservation value of individual sites based on a survey of multiple sites at one point in time, and to design conservation strategies at the site level. This was not the only possible outcome: if within-site temporal variability dominated among-site variability, among-site differences would be neither important nor predictable in the long term.

The dominance of among-site variability has important implications for conservation. There was clear evidence for the existence of a stationary distribution of long-term reef composition in East Africa. The overall shape of this distribution (Fig. 3) was similar to that estimated by Żychaluk et al. (2012) for a subset of the same data, using a different modelling approach. However, our new analysis shows that this distribution is generated by a combination of spatial and temporal processes, with substantial long-term differences among sites. Thus, the distribution in Żychaluk et al. (2012) may be a good approximation to the long-term distribution for a randomly-chosen site, but there will be much less variability over time in the distribution for any fixed site. In consequence, the sites having the highest long-term conservation value can be identified even from single-survey snapshots, and conservation strategies at the site scale may be possible. Furthermore, in cases where among-site variability in dynamics is dominant, it will be misleading to generalize from observations of a few sites to regional patterns. For example, the idea that coral reefs in the Caribbean have undergone phase shifts from coral dominance to macroalgal dominance may have been strongly influenced by observations from a small number of atypical Jamaican reefs, which do not reflect the pattern in the Greater Caribbean (Bruno et al., 2009).

In our study, the sites with the highest long-term conservation value are those with very low long-term probabilities of coral cover ≤0.1 (Fig. 5), a threshold chosen based on evidence that coral cover ≤0.1 is detrimental to reef function (Kennedy et al., 2013; Perry et al., 2013; Perry et al., 2015; Roff, Zhao & Mumby, 2015). Many of these sites are Tanzanian patch reefs, which may have maintained high coral cover despite disturbance because of local hydrography (McClanahan et al., 2007b), and are priority sites for conservation, with high alpha and beta diversity (Ateweberhan & McClanahan, 2016). Thus, it seems likely that sites of high conservation value based on community dynamics may also be sites of high diversity resulting from a combination of physical factors and biological interactions (Huston, 1985; West & Salm, 2003). However, the absence of strong spatial autocorrelation in long-term probabilities of coral cover ≤0.1 suggests that it will be necessary to consider conservation value at small spatial scales, rather than simply to identify subregions with high conservation value. Similarly, Vercelloni et al. (2014) found that trajectories of coral cover on the Great Barrier Reef were consistent at the scale of km2, but not at larger spatial scales. They argued that it would therefore be appropriate to focus management actions at the km2 scale. Also, it may be easier to persuade local communities to accept management at such scales than at larger scales (McClanahan, Muthiga & Abunge, 2016).

In this study, we aggregated all scleractinians and Millepora into a single category. Comparisons between models with this level of aggregation and models in which corals are separated into functional groups show that aggregation can hide important biological differences (e.g. Clancy et al., 2010). These differences can lead to reefs dominated by different coral taxa having very different dynamics. For example, in models of reefs in the US Virgin Islands, two sites dominated by long-lived Orbicella showed less year-to-year variability than a site dominated by short-lived species (Gross & Edmunds, 2015). These differences will affect management priorities. For example, in the Western Indian Ocean, sites which experienced a large number of Degree Heating Weeks during the 1998 bleaching event tended to show relative decreases in bleaching-susceptible genera such as Acropora and Montipora, and relative increases in resistant genera such as Porites (McClanahan et al., 2007a). Priority sites might be those in which coral diversity has been maintained through acclimation (McClanahan et al., 2007b), or in which rare and susceptible genera have survived (McClanahan et al., 2007a). Clearly, statistical models of coral dynamics would be more useful for management if they had higher taxonomic resolution. However, the number of parameters required is roughly proportional to the square of the number of taxa, except in special cases where corals are sufficiently rare that most of the interactions among taxa can be ignored (Gross & Edmunds, 2015). Separating corals into a small number of groups based on life-history strategies (Darling et al., 2012) may be the best balance between taxonomic resolution and model complexity.

Our analyses were based on the long-term consequences of current environmental conditions, and may therefore not be relevant if environmental conditions change. It is therefore better to view a stationary distribution under current conditions as a “speedometer” that tells us about the long-term outcome if these conditions were maintained, rather than as a prediction (Caswell, 2001 p. 30). Furthermore, our model did not include connectivity between sites. Although the absence of evidence for strong spatial autocorrelation in the probability of low coral cover (Section A10 and Fig. A40) suggests that connectivity is relatively unimportant for our analysis, it is possible that either current patterns of connectivity or future changes in these patterns may affect both the interpretation of stationary distributions and the optimal management strategy.

In conclusion, our analysis extends the broadly-applicable vector autoregressive approach to community dynamics (reviewed by Hampton et al., 2013) by quantifying random among-site variability in dynamics. This gives a new perspective on the long-term behaviour of the set of communities in a region, as a set of stationary distributions with random but persistent differences. The extent of these differences relative to temporal variability determines how predictable the behaviour of individual sites will be. Since these differences may be associated with differences in conservation value, probabilistic risk assessment based on this approach can be used to suggest conservation strategies at both site and regional scales. At site scales, our approach can be used to identify potential coral refugia, while at regional scales, it can identify the parameters with most influence on conservation objectives.

Supplemental Information

Supplemental Information 1 Supporting information

Click here for additional data file.

Data S1 Raw data and code

Click here for additional data file.

We are very grateful to Peter Mumby, four other anonymous reviewers and the editor for suggesting improvements to the manuscript.

Additional Information and Declarations

Competing Interests

Author Contributions

John F. Bruno is an Academic Editor for PeerJ. Tim McClanahan is an employee of the Wildlife Conservation Society.

Katherine A. Allen, Fiona Chong and Matthew Spencer conceived and designed the experiments, analyzed the data, contributed reagents/materials/analysis tools, wrote the paper, prepared figures and/or tables, reviewed drafts of the paper.

John F. Bruno conceived and designed the experiments, wrote the paper, reviewed drafts of the paper.

Damian Clancy and Kamila Żychaluk conceived and designed the experiments, analyzed the data, contributed reagents/materials/analysis tools, wrote the paper, reviewed drafts of the paper.

Tim R. McClanahan conceived and designed the experiments, performed the experiments, wrote the paper, reviewed drafts of the paper.

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
