# Peer review of "Among-site variability in the stochastic dynamics of East African coral reefs"

_PeerJ, doi:10.7717/peerj.3290_

## Round 0.1 · original submission · Major Revisions

· Academic Editor

Major Revisions

When you look at your reviews, realize that reviewers #1 and #2 are coral reef ecologists with math/quantitative skills and are, I expect, typical of readers you would like to have read and use your study. Reviewer #3 is a theoretical/quantitative biologist with strong statistical skills. This may help you refine your story for the audience you hope to reach. All reviewers gave specific advice on where improvements could be achieved.

When you return the MS, please copy each review and provide a response to each point in terms of how you dealt with that concern/suggestion, or why you did not. I won’t repeat each reviewer’s point here, but in general you should:

1) consider your intended audience and possibly do some additional word-crafting to improve communication with the less quantitative crowd,

2) address how your findings may be affected by the quality and frequency of the time series data,

3) consider clarifying Fig 3, and

4) remove the section on managing for among-site variability (see the rationale provided by reviewer #3 as 5 separate points).

I’m checking the “major revision” box on this MS due to the considerable degree of change that may be required in responding to reviewer 1 and 3, but I suspect that it will not be that difficult to successfully revise the MS. The reviewers provide good and specific guidance and I anticipate that you will be successful in a revision. So in actuality, you are somewhere between a minor and major revision. I look forward to the revised MS.

Reviewer 1 ·

Basic reporting

Overall, this is written well, but the language is highly technical and will be incomprehensible to most coral reef ecologists. The key issue here is the target audience? If this is a mathematical ecology crowd, then this could be fine (assuming it passes a math review), if it is a coral ecology/conservation ecology crowd, then more work needs to be done to make this more accessible. The problem is accentuated on page two of the manuscript with the obtuse statement "... coral cover of at least 0.1 is important". This begs the question of "0.1 of what?" - I assume it is meant to mean 10% (or is it 0.1%)? Moreover, what is meant by "coral"? Scleractinians? plus Millepora? plus octocorals?

The general ecology and coral reef literature needs to be cited more extensively and judiciously. For example:

Line 67 - the notion that coral cover of 10% (? see above) is some kind of a threshold is a very important issue and begs expansion and support.

Line 76 -- this seems like a very basic ecological observation for coral reefs that has been known for decades

Line 90 -- "undesirable community composition" is an obtuse and loaded statement that needs justification and expansion

Line 447 -- dynamics of what? This seems like a rather bland statement that states the obvious. "Where reefs do different things in different places, then you cannot generalize".

Line 465 -- this is a key finding but the value is greatly weakened by the obtuse nature of the "0.1" recommendation, the meaning of "coral", and the lack of clarity associated with "dynamics" (dynamics of what - does this refer to the speed and extent with which the community changes over time?) and "other things being equal" (what other things?).

Line 473 -- it would be helpful to address the ecological limitations of this restriction of the analysis.

Line 477 -- this is another key statement that is obtuse. It seems to imply that concentrating human activity in a specific locations is, indeed, some kind of strategy. Critically, "variability" obscures the meaning of "dynamics" (line 465 - see above). Low spatial variability in coral cover (which could include lots of sites with 0% cover) is not the same as low variability in community dynamics.

Experimental design

90% of this is a mathematical ecology paper and should be reviewed as such. I do not have the skill to evaluate this part of the paper, although it is a shame that the material is written with little attention to whether a general ecology reader or coral reef biologist can appreciate what has been done, and what implications arise from the findings.

It seems that coral ecology is heading down two roads -- one that argues for less and less resolution (pooling data to "corals" and functional groups), and one that argues for more and more resolution (splitting corals to sibling species, etc.). This paper considers "coral" which needs clarification and justification, particularly in the context of identifying refugia.

Line 119 -- if they are not "randomly chosen sites" they are not randomly chosen sites. These should really be referred to as what are -- a subset of study sites that were selected for other reasons.

Line 131-- this would benefit from taxonomic details

Validity of the findings

This needs to be reviewed by an expert with strong skills in mathematical aspects of ecology. I do not have the skills to evaluate this aspect of the work. However, the lack of general language obscures the implications of this work for most coral ecologists and conservation managers.

·

Basic reporting

There are sections described later that could be clarified.

Experimental design

One concern elaborated on below

Validity of the findings

no comment

Additional comments

This is an interesting manuscript that uses MARS models on a pretty extensive dataset of reef condition from Kenya. The key question of interest is whether the primary variance of reef state varies among reef locations or over time. The authors' assert - quite reasonably - that if much of the variance is spatial then it should be possible to identify areas that are likely to be persistently better or worse than others over time. Such patterns lend themselves to conservation decision-making. This question does not require mechanistic understanding and there is little attempt to give one.

I like the paper because the methods will be of interest to others. I have a few points that I'd like to see clarified or dealt with in a revision:

1) My own experience with MARS models is that their behaviour is very sensitive to the quality of the time series, particularly the frequency and regularity of sampling. We are given little detail here other than some sites were monitored for up to 20 years and some only twice. I'd like to see clearer methods here and an analysis of how the results were influenced by the quality of time series.

2) Figure 5 reveals that sites with a higher long-term average cover are less likely to have a cover less than 10%. Isn't this self-evident? I can't imagine how it could be any other way. In light of that, how valuable is the modelling effort? Just playing devil's advocate!

3) Some of the description of results becomes almost incomprehensible to the reader, particularly to a non-statistician. The paragraph beginning line 320 is an example. I'd like to see a clearer explanation of the results, particularly in how they relate to reefs.

Reviewer 3 ·

Basic reporting

This manuscript describes a sophisticated statistical analysis of time-series data from 20 coral reefs in Kenya and Tanzania. This is a professionally crafted manuscript that demonstrates how vector autoregressive models can be paired with long-term monitoring data to yield new insights into coral-reef dynamics. The writing is clear and succinct, the figures (for the most part) are polished and edifying, and the manuscript is well situated in the contemporary literature.

My only criticism in this category pertains to Figure 3, which struck me as a bit of a mess. While the figure provides a general impression of the predicted stationary distributions of benthic cover, any finer structure is impossible to discern. I wonder if the figure would be clearer if it were broken into four separate ternary diagrams, one for each of the four patch types.

Experimental design

High marks on all counts. The objectives are clearly stated, the core statistical methods are described cleanly and with enough detail to replicate, and the science clearly fits within the scope of the journal. I did not find any obvious mistakes in the technical development. Given the expertise of the authors, I have full confidence in the integrity of the technical results.

Validity of the findings

For the most part, but with one substantial exception, the findings are supported by the analysis. The finding that within-site variability is substantially less than among-site variability is strongly supported by the analysis, and has important implications for how we interpret variation in coral cover among sites at single snapshots in time. The majority of findings are neatly connected to and supported by the statistical analysis.

The one finding that is not supported is the suggestion that “reducing among-site variability in compositional dynamics may be an effective conservation strategy” (lines 435-6, and expounded upon in lines 465-494). I found this suggestion to be wholly without merit, for multiple reasons, which I detail below. It must be struck entirely.

The first problem with this suggestion is that it is not supported by the analysis. As I understand it, the argument seems to be that “we should minimize among-reef variability in dynamics, other things being equal … because the centre of the stationary distribution lies outside the set of compositions with coral cover <= 0.1” (lines 466-468). I do not follow the logic here. How does the position of the centre of the stationary distribution pertain to the relationship between among-site variability and the probability of regional coral cover > 10%? No quantitative argument is offered for why this should be so, which is rather jarring, following on the heels of a series of carefully reasoned quantitative arguments. If anything, the non-linear relationship between observed mean coral cover across sites and the site-level probability of coral cover < 10% (Figure 5) suggests that one cannot characterize the behavior of the ensemble of sights by merely inspecting the mean.

Second, it is entirely unclear what it means to “minimize variability in among-reef dynamics.” Taken at face value, the statement would appear to suggest that managers should intentionally remove coral from sites with above-average coral cover, so that all sites have the same coral cover. This is, of course, ridiculous.

Third, the implied logic suggests that the preferred management strategy should flip from minimizing variability to maximizing it depending on whether the centre of the stationary distribution is greater than or less than 10%. It seems self-evident that the preferred management strategy should vary smoothly with the centre of the stationary distribution, and not flip between polar opposites at the razor’s edge of a threshold value. It makes no sense that managers should prefer completely opposite strategies if the centre of the stationary distribution is 10.1% vs. 9.9%.

Fourth, although this could be easily fixed, the four key questions given at the end of the introduction make no mention of determining how among-site variability should be managed.

Fifth, the suggestion that “this result is very general” (line 470) is entirely unsupported. Merely declaring a result to be general does not make it so. No convincing quantitative argument is given for why this result should hold for these particular reefs, let alone why it should hold in general.

My language is strong here because I am concerned that many readers will skim over the details of the math, and thus will have to take the statements in the Discussion at face value. These readers will not be equipped to appreciate the distinction between the many results in the Discussion that are supported by the analysis, and this particular management suggestion, which isn’t supported by the analysis whatsoever. (And even if readers do dig into the math, only the most motivated readers are going to find section A14 and figure A36, which seem to be critical to the argument here.) As such, it is especially problematic to include this section in the manuscript.

Additional comments

All in all, this is a lovely, thoughtful, and polished manuscript, with much to commend it. (The polish and care given to the rest of the manuscript is partly what makes the portion about managing for among-site variability so startling.) The manuscript builds our statistical toolkit in a rigorous yet accessible way, and generates important insights into the dynamics at these reefs. Aside from Figure 3 and the discussion about managing for among-site variability, the manuscript is publishable as is. I congratulate the authors on an outstanding piece of work.

---

## Round 0.2 · accepted · Accept

· Academic Editor

Accept

Thank you for conducting a careful revision. The reviewers and I agree that your revision met the needs outlined by the previous reviews, so I'm happy to accept the manuscript.

Reviewer 3 ·

Basic reporting

No further comment.

Experimental design

No further comment.

Validity of the findings

No further comment.

Additional comments

This revision adequately addresses concerns raised in the first round of review. I congratulate the authors on a professionally written piece that makes an important and timely advance in our understanding of coral-reef dynamics.